

# Uniform grain-size distribution in the active layer of a shallow, gravel-bed, braided river (the Urumqi River, China) and implications for paleo-hydrology

Laure Guerit[1], Laurie Barrier[2], Youcun Liu[3], Clément Narteau[2], Eric Lajeunesse[2], Eric Gayer[2], and François Métivier[2]

[1]GET, Université de Toulouse, CNRS, IRD, UPS, Toulouse, France
[2]Institut de Physique du Globe de Paris – Sorbonne Paris Cité, Université Paris Diderot, CNRS, UMR 7154, Paris, France
[3]School of Resources and Environmental Engineering, Jiangxi University of Science and Technology, Ganzhou 341000, China

*Correspondence to:* L. Guerit (laure.guerit@get.omp.eu)

**Abstract.** The grain-size distribution of ancient alluvial systems is commonly determined from surface samples of vertically exposed sections of gravel deposits. This method relies on the hypothesis that the grain-size distribution obtained from a vertical cross-section is equivalent to that of the river bed. We report a field test of this hypothesis on samples collected on an active, gravel-bed, braided stream: the Urumqi River in China. We compare data from volumetric samples of a trench excavated in an active thread and surface counts performed on the trench vertical faces. We show that the grain-size distributions obtained from all samples are similar and that the deposit is uniform at the scale of the river active layer, a layer extending from the surface to a depth of approximately ten times the size of the largest clasts.

*Copyright statement.* TEXT

## 1 Introduction

The size of the river-bed sediments and its spatial distribution result from transport and deposition mechanisms in alluvial systems. These mechanisms have been intensively studied to model fluvial behavior and landscape evolution (Wilcock and McArdell, 1993; Paola and Seal, 1995; Vericat et al., 2008; Piedra et al., 2012; Sun et al., 2015), and the temporal variations in grain-size distributions can be used to reconstruct paleo-environments or changes in tectonic and climatic conditions (Duller et al., 2010; Whittaker et al., 2011; Michael et al., 2014; Schlunegger and Norton, 2015; D'Arcy et al., 2016; Chen et al., Acc.). For this, the granulometry in ancient systems is often characterized on the basis of a single grain-size distribution sampled along vertical conglomeratic outcrops with a limited extension (Duller et al., 2010; Whittaker et al., 2011; Chen et al., Acc.). This simplification can be relevant to derive quantitative information about a stream from its deposits, but it is based on two assumptions. First, the sampled deposits must be equivalent to the ones that were in direct contact with the flow, and thus





actively involved in the transport and deposition processes. Second, the grain-size distribution obtained from vertical outcrops must be equivalent to the reach-scale distribution (i.e. to the granulometry of the whole river bed).

In alluvial gravel systems, sediments can experience strong spatial variations in size. Downstream fining, which results from abrasion and preferential deposition of coarse particles, dominates the large-scale evolution of granulometry along a river path

(Parker, 1991; Paola and Seal, 1995; Singer, 2008; Rice and Church, 2010). This preferential deposition of the coarse grains, together with the removal of fine grains by winnowing during low flows, can be responsible for the formation of a coarse layer, or armour, at the surface of the bed (Parker and Klingeman, 1982; Church et al., 1987; Wilcock and McArdell, 1993; Mao et al., 2011). This layer generally forms at reach scale although it is not always observed (Laronne et al., 1994; Laronne and Shlomi, 2007; Storz-Peretz and Laronne, 2013b, 2018). In addition, local (from a meter to a few tens of meters) trends in grain

size are observed at the scale of the morpho-sedimentary elements (e.g., bars, anabranches, and chutes for the braided rivers) that built the river bed with coarser or finer grained patches (Fig. 1)(Bluck, 1971; Smith, 1974; Milne, 1982; Lisle and Madej, 1992; Ashworth et al., 1992; Laronne et al., 1994; Laronne and Shlomi, 2007; Guerit et al., 2014; Storz-Peretz et al., 2016). As a consequence, two different features of granulometric sorting can occur at a given location along a stream: first, a vertical evolution at reach scale with a surface layer (the first centimeters of the bed deposits, often scaled with the larger grains)

coarser than what is below (the subsurface layer), and second, local lateral variations associated with the morpho-sedimentary elements.

However, recent experimental findings suggest that the granulometry of gravel-bed braided streams might be uniform above a given scale. Indeed, over a hydrological season, grains actively involved in transport and deposition are contained within the active layer of a river bed. Using a physical model of braided streams, Leduc et al. (2015) show that this layer extends

laterally over the whole river bed and that its thickness corresponds to the maximum difference in bed elevation measured on the surveyed reach. They also observe that the active layer scales with the largest clast of the bed, and extends over a thickness closed to 10 $D_{90}$ (the $90^{th}$ percentile of the grain size distribution). Based on the spatial organization of deposits with different calibers, a few experimental studies show that the sediments are well-mixed in this active layer and thus suggest that the grain-size distribution of gravel-bed braided rivers is uniform at the scale of the active layer (Gardner and Ashmore, 2011; Leduc

et al., 2015; Gardner et al., 2018) . However, such an analysis has not yet been performed on natural rivers. The first aim of this study is therefore to investigate the granulometric uniformity of the active layer of a gravel-bed braided river, which is a prerequisite for a relevant paleo-grain size sampling.

Two methods are commonly used to characterize the granulometry of gravel deposits (grains larger than 4 mm, Wentworth, 1922): the surface count (grid-by-number) and the volumetric (sieve-by-weight) methods. The first one consists in measuring

the intermediate axis ($b$-axis) of the grains lying on the top of a river bed and located at the nodes of a predefined grid. It is classically used to determine the surface granulometry of present-day stream beds (Wolman, 1954; Church et al., 1987; Bunte and Abt, 2001). The second one consists in sieving a volume of sediments excavated from a river bed. It is generally used to sample the subsurface or bulk granulometry. The grain-size distribution obtained by this method is generally considered as representative of the whole river bed (Church et al., 1987; Bunte and Abt, 2001). The two methods lead to grain-size

distributions that can be directly compared (Kellerhalls and Bray, 1971; Church et al., 1987; Bunte and Abt, 2001). In ancient



alluvial systems, sediments are often cemented and it is not always possible to remove grains from outcrops. Photographic approaches, that do not require grain extraction, can be implemented for such outcrops but these methods suffer from the 2D exposure of 3D grains, which biases the measurement of their diameter (Kellerhalls and Bray, 1971; Church et al., 1987; Diplas and Fripp, 1992; Bunte and Abt, 2001; Storz-Peretz and Laronne, 2013a; Buscombe, 2013). Consequently, Wolman's

methodology adapted on vertical sections is preferably used (Duller et al., 2010; Whittaker et al., 2011; Michael et al., 2014; D'Arcy et al., 2016; Chen et al., Acc.). However, this method has been developed to characterize the surface granulometry of active rivers where grains can be easily picked up and measured. To date, its validity on vertical sections has not been demonstrated. This is the second aim of this study.

In this article, we present a granulometric study on the Urumqi River, an active braided river in China. First, we describe

the methodology implemented to sample the grain-size distributions by horizontal or vertical surface counts and volumetric sieving. Then, based on a large data set, we show that despite local heterogeneities probably associated with the morpho-sedimentary elements of the river bed, the grain-size distribution of the sediments is uniform at the scale of the active layer. In addition, we observe that the distribution obtained by vertical surface counts is similar to the bulk granulometry of the river bed. This study thus shows that quantitative information about the granulometry of paleo-rivers can be accurately derived from

surface counts along vertical conglomeratic outcrops.

## 2  Sampling site

The Urumqi River is a shallow (<1 m-deep), gravel-bed, braided river draining the northern side of the Tian Shan Range in China (Figs. 1 and 2a-b). The river initiates at the front of a glacier in the high range at 3600 m and runs northwards to the Junggar Basin where it dies out into the desert at an elevation of 1100 m. The sediments sampled in this study are located about

10 km downstream of the mountain topographic front. There, the Urumqi River braids within an alluvial valley cut into the deposits of a Pleistocene alluvial fan (Zhou et al., 2002; Guerit et al., 2016) (Fig. 2a-b). At this location, the catchment area of the river is close to 1000 km$^2$, its average slope is 0.02, and its runoff is mainly due to summer rains and snow or ice melt. As a consequence, the river mostly flows from May to September with a mean annual discharge of about 7.5 m$^3$ s$^{-1}$ (Zhou et al., 1999) (Fig. 2b-c) and a total sediment load of 1-2 10$^8$ kg yr$^{-1}$ (Liu et al., 2008, 2011). The river is almost dry outside of the

high flow season and we can thus measure the granulometry of its bed (Fig. 1). The sediments found at the surface of the river bed are mostly gravels (Guerit et al., 2014).

To estimate the thickness of the active layer of the Urumqi River, we acquired 5 transverse topographic profiles across the river bed with a Timble S6 DR300+ total station with a point every meter on average (Fig. 3). Differences in elevation between the highs and the lows of the river bed are in the order of 1 meter (Fig. 3). We thus estimate the active layer of the Urumqi River

to be ∼1 meter. This morphological estimate is in good agreement with the one based on the $D_{90}$ of the grain-size distribution at the same location. Indeed, the $D_{90}$ is ∼ 10 cm at the sampling site (Guerit et al., 2014) and according to this value, the active layer should also extend from the surface down to 1 meter (i.e. 10 $D_{90}$) (Leduc et al., 2015).





## 3 Methodology

We combine two methods to characterize the grain-size distribution of the Urumqi River: the surface count and the volumetric methods.

To characterize the surface granulometry of the river bed at reach scale, we perform an horizontal surface count over the
whole river width (Fig. 4a). The grid is positioned perpendicular to the main water flow direction, with nodes every 10 m. We choose the spacing of the grid larger than the size of the granulometric patchiness related to the river morpho-sedimentary elements (Guerit et al., 2014). We extract the grains located directly under each node and measure their $b$-axis. However, grains smaller than 4 mm (36% of the total sample) are not considered in the analysis in order to reduce the measurement uncertainties (Kellerhalls and Bray, 1971; Church et al., 1987; Bunte and Abt, 2001). The resulting sample is composed of 351
grains with $D \geq 4$ mm. The uncertainties associated with the grain-size distribution obtained by this methodology are mainly due to the limited number of measurements. It is generally considered that 100 grains must be measured in order to accurately characterize the $D_{50}$ of a distribution with a surface count, while 400 grains are required for the coarse quantile $D_{90}$ (Wolman, 1954; Church et al., 1987; Rice and Church, 1996). In this study, we evaluate the uncertainties on these characteristic diameters by a bootstrapping method (Bunte and Abt, 2001). We estimate the quantiles of 10530 distributions built by randomly sampling
with replacement 1 to 351 grains from our surface sample. We find the $D_{50}$ and $D_{90}$ to be defined within a range of $\pm 15\%$ (Fig. 5a).

To characterize the surface-layer and subsurface granulometries of the river bed, we dig into its deposits a $7.2 \times 1.2 \times 1$ m trench perpendicular to the flow direction of one thread (Fig. 4b-c). The sediments are excavated step by step from this trench as individual volumetric samples. We set the thickness of these samples to 2 $D_{90}$ to insure that the largest grains are contained
within one sample, and we determine the volume to be large enough to accurately characterize the grain-size distribution. Indeed, the accuracy of the volumetric method depends upon the sample weight with respect to the weight of its largest clast (Church et al., 1987; Haschenburger et al., 2007). Ideally, the largest grain of a volumetric sample should not contribute to more than 0.1% of the total weight. In this case, the grain-size fractions are determined with a 0.1% precision. However, this criterion is difficult to achieve on the field and in this study, the fractions are defined within $\pm$ 5%. The corresponding individual
volumetric samples are $1.2 \times 1.2 \times 0.2$ m, resulting in 30 samples labelled from 1 to 5 with respect to their depth, and from A to F with respect to their lateral position (Fig. 4c). We sieve the sediments using mesh sizes ranging from 25.6 cm down to 63 $\mu$m. We then weight the grains retained in each mesh to obtain a mass for a given diameter, which is considered to be the $b$-axis of the clasts (Church et al., 1987; Bunte and Abt, 2001; Guerit et al., 2014). To be consistent with the surface counts, for which only the grains larger than 4 mm are considered, we remove all the grains smaller than 4 mm (24% of the volumetric
samples on average) from the analysis. These individual volumetric samples are combined in different ways. (i) They are used individually to characterize the grain-size distribution at local scale. (ii) They are merged according to their depth (layers 1 to 5) or to their lateral position (columns A to F) to document potential granulometric variability associated with the location of a sample within the river bed. (iii) These individual volumetric samples are also merged altogether to characterize the bulk





granulometry of the Urumqi river bed at the scale of the active layer. (iv) Finally, they are merged randomly and analyzed to document the granulometric evolution with respect to the weight of a sample.

In order to test whether the bulk granulometry of the river bed can be characterized from the sampling of a cross-section, we eventually determine the grain-size distribution of the sediments outcropping on the walls of the trench by a vertical surface

count. We implement the Wolman methodology on the vertical sections of the river deposits (Fig. 4d), using a square grid of 20 cm ($\sim 2 \, D_{90}$ in order to avoid sampling twice the same grain). We extract and measure the $b$-axis of the grains under each node. As for the horizontal surface count, the smallest grains (19% of the total sample) are not considered and the resulting sample is composed of 298 grains with $D > 4$mm. To evaluate the uncertainties associated with the main diameters, we estimate the quantiles of 8940 distributions built by randomly sampling with replacement 1 to 298 grains from our vertical count sample

(Bunte and Abt, 2001). We find the $D_{50}$ and $D_{90}$ to be defined within a range of $\pm 15\%$ and $\pm 20\%$, respectively (Fig. 5b).

## 4   Results

### 4.1   Granulometry of the river bed at different spatial scales

#### 4.1.1   Grain-size distribution of volumetric samples

First, we analyze the grain-size distributions of the individual volumes excavated from the trench (Fig. 6a, Table 1). These 30

volumetric samples have an average weight of 440 kg and they are composed by more than 85% of pebbles (i.e. grains with $D \in$ 4-64 mm). Their median diameter $D_{50}$ ranges between $17\pm1$ mm and $32\pm2$ mm, while their $D_{90}$ is comprised between $52\pm3$ and $126\pm6$ mm. Although the 30 distributions are similar in shape, we observe some scatter between them (Fig. 6a, Table 1). In particular, the $D_{50}$ and the $D_{90}$ vary within a range of $\pm$ 25% and $\pm30\%$ around the means of the samples (23 and 73 mm, respectively).

#### 4.1.2   Vertical sorting

Second, we merge the individual volumetric samples according to their depth (layers 1 to 5) and analyze the grain size sorting with respect to depth (Fig. 6b, Table 2). On average, the different layers weight 2600 kg and they are composed by 93% of pebbles. Their $D_{50}$ ranges between $21\pm1$ and $25\pm1$ mm, while their $D_{90}$ is comprised between $65\pm1$ and $76\pm2$ mm. Accordingly, the granulometries of these five layers show a limited scattering, with the $D_{50}$ and $D_{90}$ varying within a range

of $\pm9\%$ around the means of the samples (23 and 70 mm, respectively). The grain-size distributions of these larger samples is less scattered than the individual ones. In addition, the ratio between the $D_{50}$ of the surface layer (i.e. layer 1 between 0 and 0.2 m, ) and the sub-surface ones (i.e. layers 2 to 5 between 0.2 and 1 m) is of 1.07. The surface layer of the deposits is thus indistinguishable from the subsurface. Therefore, at the scale of the active layer, there is no vertical sorting in the Urumqi river bed.



### 4.1.3 Horizontal sorting

Third, we analyze the grain-size distributions of the individual volumes merged according to their lateral position (columns A to F) (Fig. 6c, Table 3). On average, the different columns weight 2200 kg and they are composed by 92% of pebbles. Their $D_{50}$ ranges between 21±1 and 26±1 mm, while their $D_{90}$ varies between 64±1 and 76±2 mm. Here again, the granulometric

distributions exhibit limited scattering as the $D_{50}$ and $D_{90}$ vary within a range of ±13% and ±10% around the means of the samples (23 and 71 mm, respectively). Thus, at the scale of the active layer, the Urumqi river bed has no horizontal grain-size sorting either. In consequence, grain-size distributions issued from vertical pits are equivalent to distributions issued from horizontal layers.

### 4.1.4 Volumetric versus surface grain-size distributions

Finally, we merge all the volumetric samples together to determine the bulk granulometry of the river bed. Based on 13150 kg of sediments, this bulk volumetric distribution is composed of 92% of pebbles. It has a $D_{50}$ of 23±1 mm and a $D_{90}$ of 73±1 (Fig. 6d, Table 4). This is in agreement with the reach-scale grain-size distribution characterized by surface counts. Indeed, based on 351 grains ($D \geq 4$ mm), the surface distribution is made up of 78% of pebbles. Its median diameter $D_{50}$ is 30±5 mm, while the $D_{90}$ is 100±15 mm (Fig. 6d, Table 4). The surface granulometry is thus slightly coarser than the bulk one, but

no major shift, depletion or enrichment in the fine or coarse fractions is observed between the two distributions.

### 4.1.5 Armouring versus uniformity

The uniformity in the grain-size distributions of the various samples we analyzed (individual volumes, layer and column samples), and their similarity with the bulk one, suggest that the Urumqi river bed is not armoured at reach scale. In addition, the absence of vertical or lateral sorting in grain size within the active layer implies that at any depth or location of this layer,

the deposits are representative of the sediments transported as bedload in direct contact with the flow. Our results therefore accord with the experimental findings of Gardner and Ashmore (2011), Leduc et al. (2015) and Gardner et al. (2018).

### 4.2 Granulometric uniformity of the active layer

Although the grain-size distributions of the different samples resemble the bulk one, we observe some scatter that seems to be dependent on the sample size. Indeed, the grain-size distributions of the 30 individual volumetric samples are more

scattered than the distributions of the layer and column samples. To assess the minimum weight required for the grain-size distributions to converge toward a sample equivalent to the bulk one, we randomly merge with replacement the individual samples to determine the $D_{50}$ and $D_{90}$ as a function of the sample weight (Fig. 7). By authorizing replacement, we maximize the variability of the virtual samples. We observe that variability around the mean decreases with increasing weight, and at first order, sample weight must be multiplied by a factor 2 to decrease the variability by 5%. Accordingly, the bulk grain-size

distribution can be determined within a range of ±10% from a sample composed of 4000 kgs of sediments (∼2 m³) larger than 4 mm (Fig. 7), but we observe that $> 10000$ kg of sediments are required to compose a sample equivalent to the whole trench




one. In fact, above this weight, both the $D_{50}$ and the $D_{90}$ are equivalent to the ones of the whole trench ($23\pm1$ and $73\pm4$ mm, respectively). At the scale of the active layer, the grain-size distribution of the Urumqi River deposits can thus be considered as uniform, providing that the sample size excesses the typical size of the morpho-sedimentary elements of the river bed.

### 4.3   Equivalence of sampling methods

Finally, we analyze the grain-size distribution obtained by vertical counts along the surface of the walls of the trench (Fig. 6d, Table 4). Based on 298 grains ($D \geq 4$ mm), this sample is made up of 85% of pebbles. Its median diameter $D_{50}$ is $20\pm4$ mm while its $D_{90}$ is $82\pm16$ mm. This grain-size distribution compares well with the reach-scale surface count and the bulk volumetric distribution of the trench (Fig. 6d). This comparison indicates that the characterization in cross-section of the Urumqi active layer is equivalent to the more traditional horizontal surface count and volumetric methods. The similarity between the distributions documented by these different sampling methods thus shows for the first time that surface counts implemented on vertical outcrops can be used as proxies for a bulk volumetric sampling.

### 5   Discussion

### 5.1   Grain-size sorting

At local scale ($<$ meter to 10s of meters), features of grain-size sorting are commonly observed and documented on the bed of braided gravel-bed streams (Leopold et al., 1964; Bluck, 1971, 1976; Smith, 1974; Ashworth, 1996; Ashmore, 2013; Guerit et al., 2014). In these rivers, this surface sorting is associated with the morpho-sedimentary elements (bars, anabranches and chutes) that shape the river bed. In the Urumqi River, local variations in grain sizes are also visible in the vertical section of the active layer with small areas enriched in fine or in coarse grains (Fig. 8). This variability could be the expression in cross-section of the small-scale sorting associated with the morpho-sedimentary elements observed on the bed (Guerit et al., 2014). This sorting may be responsible for the scatter observed between the granulometric distributions of the 30 individual volumetric samples (Fig. 6a) that scale with the morpho-sedimentary elements. This is supported by the decreasing scatter between the grain-size distributions with increasing sample size (Figs. 6 and 7). In particular, at the sampling site, a grain-size uniformity is observed above a given amount of sediments. Based on the virtual distribution built by random merging of the sample, 4000 kgs (corresponding to $\sim$2 m$^3$) seems to be required (Fig. 7), but the granulometric similarity between the layers, or between the columns, suggests that uniformity may arise for a smaller amount of sediments (Fig. 6b and c). The active layer of the Urumqi River thus appears as the superposition of small-scale structures (the morpho-sedimentary elements), whose variability vanishes above a given scale. Accordingly, at the scale of the active layer, the deposits of the Urumqi River are uniform in space in terms of grain-size distribution. In particular, no trend of lateral or vertical sorting is observed within the active layer of the river bed (Fig. 6) in agreement with previous experimental works (Gardner and Ashmore, 2011; Leduc et al., 2015; Gardner et al., 2018). The agreement between field and physical experiments lead us to propose that the absence of vertical and lateral sorting may be typical of non-armoured, gravel-bed braided rivers.




## 5.2 Vertical sampling

Finally, the granulometric uniformity at the scale of the active layer implies that the grain-size distribution of paleo-rivers can be adequately determined from conglomeratic outcrops. In fact, the calibers of the river deposits are similar to the ones of the grains directly in contact with the flow. We show that the broadly used Wolman grid-by-number method can be implemented on vertical sections to characterize grain-size distributions with a good level of confidence. Our study thus legitimates this acquisition, for braided rivers at least, for stratigraphical or paleo-hydrological reconstructions (Duller et al., 2010; Whittaker et al., 2011; Michael et al., 2014; D'Arcy et al., 2016; Chen et al., Acc.). However, two limitations must be considered to generalize our results to any field work. First, this study is based on the grains removed from the trench walls so that it is possible to measure their actual $b$-axis. In ancient systems, deposits are often cemented and it is not always possible to remove grains from the outcrop. In that case, outcropping diameters should be identified before implementing measurements. Indeed, in a section perpendicular to the main flow, the $b$- (intermediate) and $c$- (small) axis are expected to be visible. In such situation, the $b$-axis will appear to be the longest one. On the contrary, in a section parallel to the main flow, the $a$- (long) and $b$- (intermediate) axis are expected to be visible, and the $b$-axis will then appear to be the shortest one (e.g., Bunte and Abt, 2001). Indications of paleo-flows can help to recognize the actual $b$-axis. The second limitation is related to the thickness of the active layer, which extends in depth over several $D_{90}$ only (Laronne et al., 1994; Gardner and Ashmore, 2011; Leduc et al., 2015). In addition, in stratigraphic successions, deposits are stacked vertically through time with potential changes of the river characteristics and preservation rate during the sedimentation. Consequently, to compensate for the limited extension in depth of the active layer and for the potential sedimentation changes through time, the grids used for vertical surface counts should extend laterally rather than vertically to stay as much as possible within the same sedimentary layer.

## 6 Conclusions

We perform a granulometric study on the deposits of the Urumqi River, an active, gravel-bed, braided stream in China. Based on a large data set collected by surface counts and volumetric samplings, we show that the grain-size distribution of the river bed is uniform at the scale of its active layer. Despite some local variabilities, there is no vertical or lateral granulometric trend within this layer. Because our findings confirm earlier physical models (Gardner and Ashmore, 2011; Leduc et al., 2015; Gardner et al., 2018), we propose that, beyond the Urumqi River, this uniformity be the case for all the non-armoured, gravel-bed braided streams.

This uniformity implies that it is possible to determine the grain-size distribution of gravel-bed braided alluvial systems from vertical outcrops. We show that the grid-by-number method, initially developed and tested on the horizontal surface of river beds, can be implemented on vertical outcrops to obtain samples equivalent to a volumetric investigation. This study thus brings support to the hypothesis that vertical surface counts provide an accurate characterization of the grain-size distribution of paleo-braided rivers.



*Competing interests.*  TEXT

The authors declare no competing interests.

*Acknowledgements.*  The study presented in this manuscript was conducted with research grants from the CNRS-INSU RELIEF program and the French-Chinese CNRS-CAS International Associated Laboratory SALADYN, as well as from the IPGP BQR and Potamology programs.

5   L.G. benefitted from a PhD grant from the French Ministry of Research and Higher Education (MESR). This work is partially supported by the National Natural Science Foundation of China (No. 41471001).



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




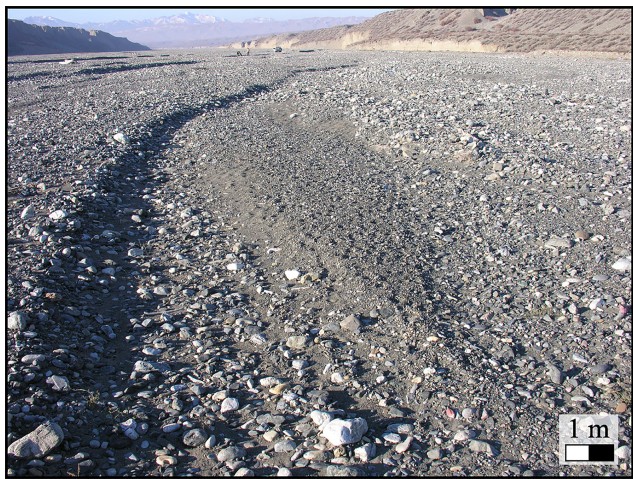

**Figure 1.** Example of a shallow gravel-bed braided river bed during the dry season (Urumqi River, China). Spatial variations in grain size can be observed at the surface of the river bed with finer- and coarser-grained areas.

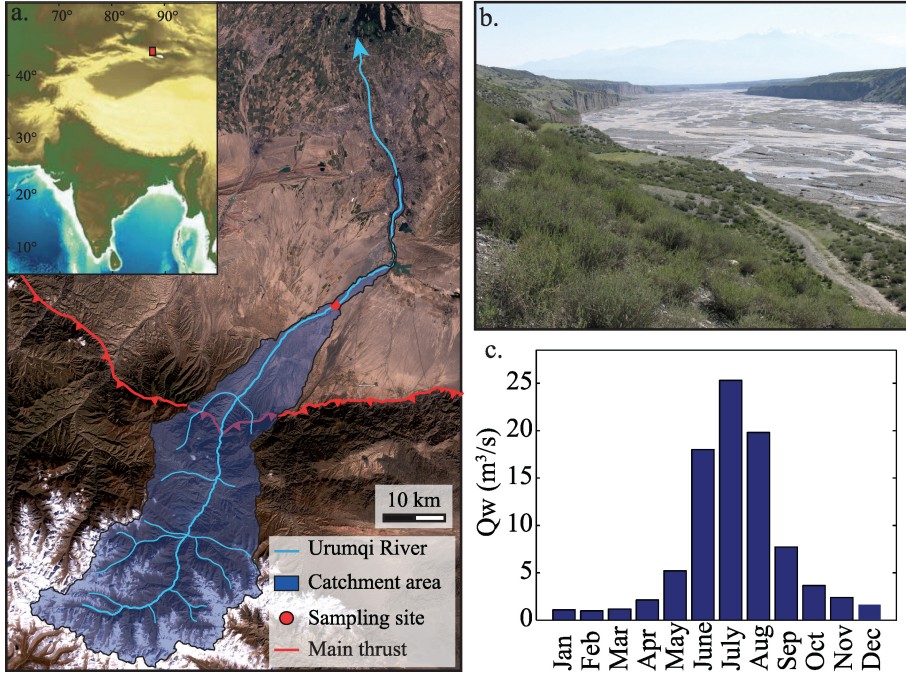

**Figure 2.** a) Location, drainage network and catchment area of the Urumqi River system, b) picture of the river at the sampling site during the high flow season, and c) annual hydrograph of the river (after Zhou et al., 1999).





**Figure 3.** Transverse topographic profiles acquired along the Urumqi River bed.

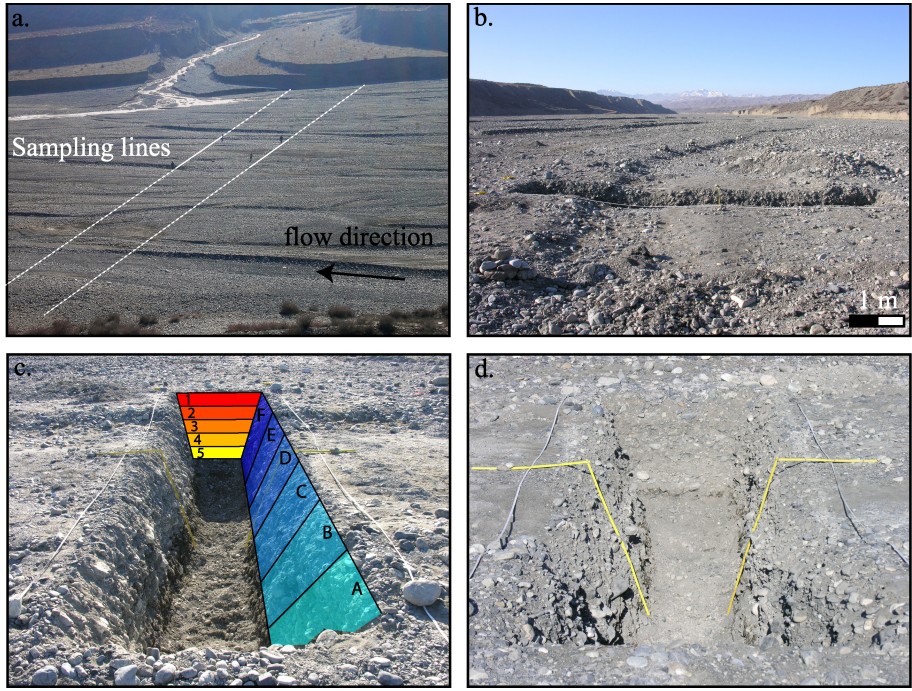

**Figure 4.** a) Implementation of the horizontal surface counts at reach scale with a node every 10 m. View b) from the north and c) from the west of the trench. On this second view, the sampling nomenclature is indicated. Layers are labelled from 1 to 5 and colored from red to yellow, whereas columns are labelled from A to F and colored from light to dark blue. d) Implementation of the vertical surface counts along the trench walls with a node every 20 cm.




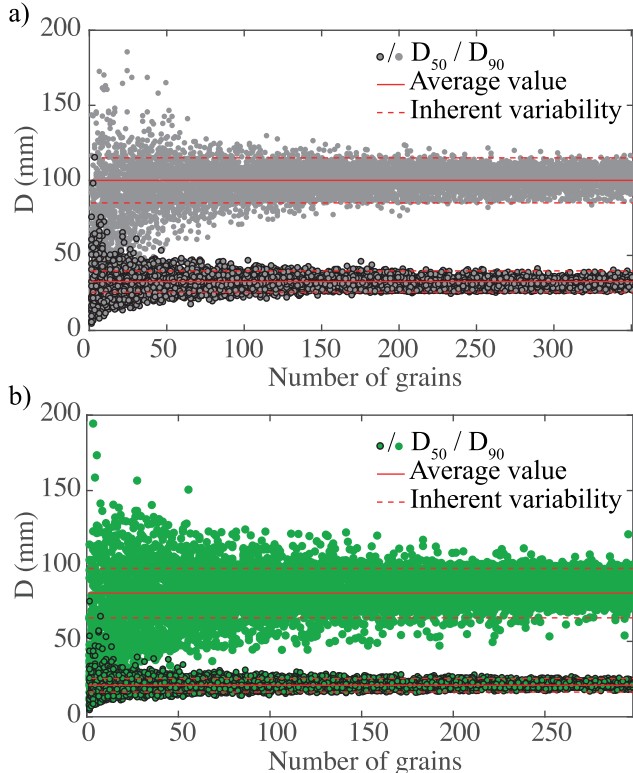

**Figure 5.** Uncertainties associated with the $D_{50}$ and $D_{90}$ obtained by surface counts a) at reach scale and b) along the walls of the trench. Diameters are defined within a range of $\pm$15-20%.





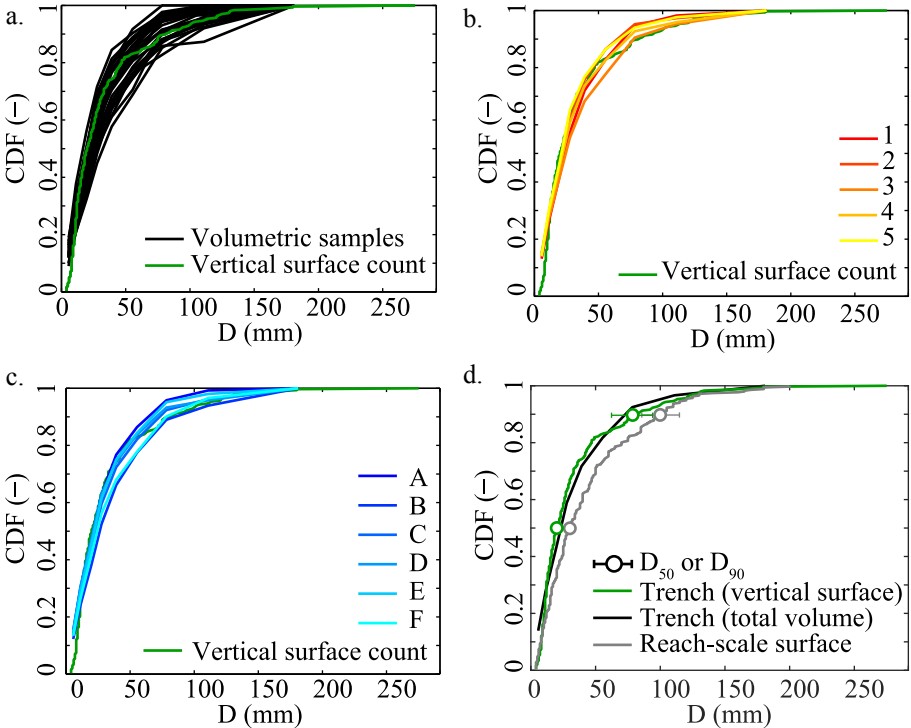

**Figure 6.** Grain-size distributions of a) the 30 individual volumetric samples excavated from the trench (1.2×1.2×0.2 m, black lines), b) the five 20 cm-thick layers of the trench from the surface to 1 m-deep (red to yellow), and c) the six 1.2 m-wide columns of the trench from the west to the east (dark to light blue). On each figure, the granulometry sampled by vertical surface counts on the trench wall is indicated (green line). d) Grain-size distributions of the three bulk samples obtained by horizontal surface counts at the scale of the reach (gray), volumetric sieving of the trench (black), and vertical surface counts on the walls of the trench (green). These four panels suggest that the sediments of the Urumqi River active layer are uniform at the scale of the active layer in terms of grain-size distribution.





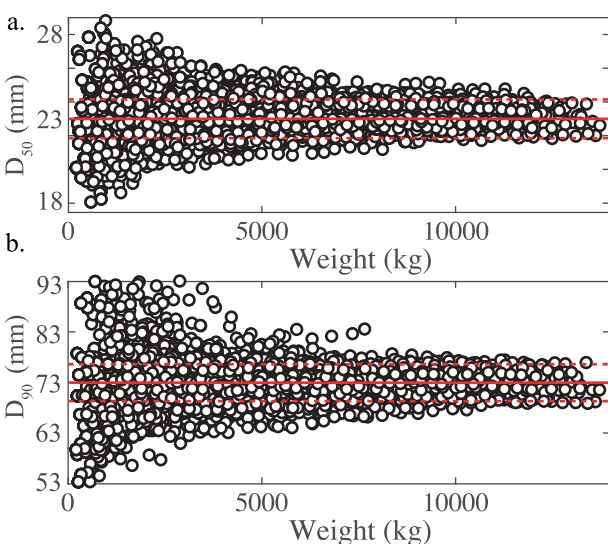

**Figure 7.** Evolution of a) the $D_{50}$ and b) the $D_{90}$ with respect to the sample weight. The grain-size distributions corresponding to the quantiles shown on this graph are built by random merging of the individual volumetric samples. Red lines are for the mean value, dotted lines indicate $\pm 5\%$.



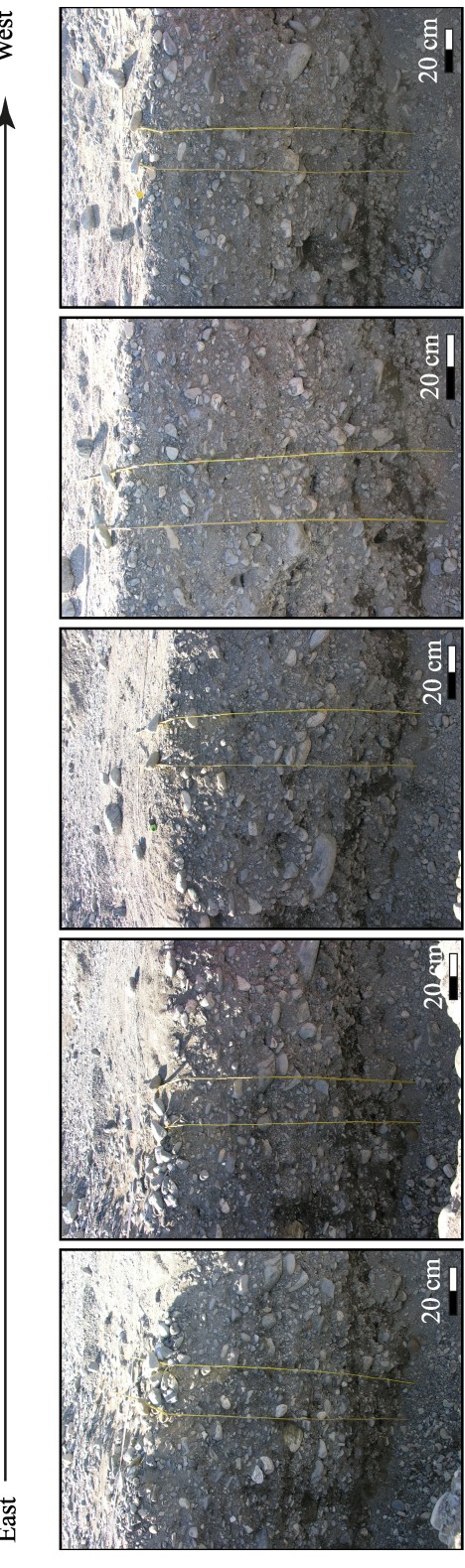

**Figure 8.** Photographies of the trench walls from east to west. Local heterogeneities with more fine or coarse grains can be observed but at the scale of the active layer, the sediments show no vertical or lateral stratification.



**Table 1.** Main characteristics of the local-scale samples excavated from the trench. $P$ and $C$ are the proportions of pebbles and cobbles within the sediments defined after Wentworth (1922). $D_{50}$ is the mean diameter and $D_{90}$ the $90^{th}$ quantile of the grain-size distributions. Confidence intervals are calculated from the Church et al. (1987)'s criteria.

| Sample | Size | P (%) | C (%) | $D_{50}$ (mm) | $D_{90}$ (mm) | Sample | Size | P (%) | C (%) | $D_{50}$ (mm) | $D_{90}$ (mm) |
|--------|------|-------|-------|---------------|---------------|--------|------|-------|-------|---------------|---------------|
| A1 | 649 kg | 96 | 4 | 23 ± 1 | 66 ±3 | D1 | 372 kg | 97 | 3 | 17 ± 1 | 52 ±3 |
| A2 | 300 kg | 100 | 0 | 21 ± 1 | 56 ±3 | D2 | 309 kg | 92 | 8 | 22 ± 1 | 74 ±4 |
| A3 | 506 kg | 96 | 4 | 20 ±1 | 62 ±3 | D3 | 273 kg | 95 | 5 | 23 ± 1 | 70 ±4 |
| A4 | 475 kg | 93 | 7 | 20 ±1 | 70 ±4 | D4 | 314 kg | 86 | 14 | 26 ± 1 | 126 ±6 |
| A5 | 338 kg | 89 | 11 | 22 ±1 | 66 ±3 | D5 | 241 kg | 97 | 3 | 20 ± 1 | 60 ±3 |
| B1 | 607 kg | 85 | 15 | 32 ±2 | 97 ±5 | E1 | 618 kg | 97 | 3 | 25 ± 1 | 65 ±3 |
| B2 | 615 kg | 95 | 5 | 24 ±1 | 65 ±3 | E2 | 465 kg | 97 | 3 | 21 ± 1 | 60 ±3 |
| B3 | 599 kg | 88 | 12 | 29 ±1 | 94 ±5 | E3 | 504 kg | 88 | 12 | 25 ± 1 | 78 ±4 |
| B4 | 510 kg | 88 | 12 | 22 ±1 | 74 ±4 | E4 | 441 kg | 97 | 3 | 21 ± 1 | 64 ±3 |
| B5 | 343 kg | 83 | 17 | 27 ±1 | 104 ±5 | E5 | 484 kg | 96 | 4 | 20 ±1 | 60 ±3 |
| C1 | 363 kg | 97 | 3 | 20 ± 1 | 61 ±3 | F1 | 537 kg | 88 | 12 | 24 ±1 | 88 ± 4 |
| C2 | 368 kg | 88 | 12 | 24 ± 1 | 78 ±4 | F2 | 554 kg | 91 | 9 | 27 ±1 | 77 ± 4 |
| C3 | 271 kg | 90 | 10 | 27 ± 1 | 77 ±4 | F3 | 399 kg | 91 | 9 | 23 ±1 | 70 ± 4 |
| C4 | 386 kg | 90 | 10 | 25 ± 1 | 76 ±4 | F4 | 525 kg | 87 | 13 | 25 ±1 | 89 ±4 |
| C5 | 271 kg | 98 | 2 | 20 ± 1 | 60 ±3 | F5 | 509 kg | 91 | 9 | 23 ±1 | 73 ±4 |

**Table 2.** Main characteristics of the five layers issued from the trench. $P$ and $C$ are the proportions of pebbles and cobbles within the sediments defined after Wentworth (1922). $D_{50}$ is the mean diameter and $D_{90}$ the $90^{th}$ quantile of the grain-size distributions. Confidence intervals are calculated from the Church et al. (1987)'s criteria.

| Sample | Size | P (%) | C (%) | $D_{50}$ (mm) | $D_{90}$ (mm) |
|--------|------|-------|-------|---------------|---------------|
| Layer 1 | 3226 kg | 95 | 5 | 24 ±1 | 69 ±1 |
| Layer 2 | 2523 kg | 95 | 5 | 22 ±1 | 65 ±1 |
| Layer 3 | 2657 kg | 91 | 9 | 25 ±1 | 76 ±2 |
| Layer 4 | 2566 kg | 92 | 8 | 22 ±1 | 72 ±1 |
| Layer 5 | 2161 kg | 93 | 7 | 21 ±1 | 66 ±1 |
| Average | 2626 kg | 93 | 7 | 23 ±1 | 70 ± 1 |





**Table 3.** Main characteristics of the six columns issued from the trench. $P$ and $C$ are the proportions of pebbles and cobbles within the sediments defined after Wentworth (1922). $D_{50}$ is the mean diameter and $D_{90}$ the $90^{th}$ quantile of the grain-size distributions. Confidence intervals are calculated from the Church et al. (1987)'s criteria.

| Sample | Size | P (%) | C (%) | $D_{50}$ (mm) | $D_{90}$ (mm) |
|---|---|---|---|---|---|
| Column A | 2268 kg | 95 | 5 | 21 ±1 | 64 ±1 |
| Column B | 2674 kg | 88 | 12 | 26 ± 1 | 77 ± 2 |
| Column C | 1659 kg | 92 | 8 | 23 ±1 | 73 ± 1 |
| Column D | 1509 kg | 93 | 7 | 21 ±1 | 71 ± 1 |
| Column E | 2512 kg | 95 | 5 | 22 ±1 | 67 ± 1 |
| Column F | 2524 kg | 90 | 10 | 24 ±1 | 76 ± 2 |
| Average | 2191 kg | 92 | 8 | 23 ±1 | 71 ±1 |

**Table 4.** Main characteristics of the large-scale samples. $P$ and $C$ are the proportions of pebbles and cobbles within the sediments defined after Wentworth (1922). $D_{50}$ is the mean diameter and $D_{90}$ the $90^{th}$ quantile of the grain-size distributions. Confidence intervals are calculated by bootstrapping for the surface counts and from the Church et al. (1987)'s criteria for the volumetric sample.

| Sample | Size | P (%) | C (%) | $D_{50}$ (mm) | $D_{90}$ (mm) |
|---|---|---|---|---|---|
| Horizontal surface count | 351 grains | 78 | 22 | 30 ± 5 | 100 ± 15 |
| Total volume | 13150 kg | 92 | 8 | 23 ±1 | 73 ±4 |
| Vertical surface count | 298 grains | 85 | 15 | 20 ± 4 | 82 ± 16 |