# Peer review of "Uniform grain-size distribution in the active layer of a shallow, gravel-bed, braided river (the Urumqi River, China) and implications for paleo-hydrology"

_Earth Surface Dynamics, 2018_

## Referee Comment (RC1) · Anonymous Referee #1 · 6 Jul 2018

Guerit et al. present a field study from the Urumqi River, China in which they compare different methods of grain-size analysis including (i) horizontal surface counts over the whole river width, (ii) vertical surface counts on an outcropping trench wall and (iii) volumetric counts (sieving) of a 1 m deep trench excavated within the dry channel-bed. As they found no differences in sub-sample grain-size distributions in vertical nor horizontal direction within the trench, they propose that the grain-size distribution is uniform within the active layer, which might be a typical phenomenon for non-armoured gravel-bed, braided rivers. Second, they found no difference between the volumetric

grain-size analysis and the vertical surface counts within the same trench. They conclude that the surface point count method, which was originally developed by Wolman for horizontal surface granulometry analyses in active rivers, can also be applied to vertical outcrops.

Temporal variations in grain-size distribution are used to reconstruct paleo environmental conditions including climate and tectonics. As such, it is important to investigate the differences between methods that are commonly applied to characterize grain-size distributions. As this study performs a very systematic comparison of three of those methods in a natural, gravel-bed, braided river, and we generally lack those systematic method validations, the study is a valuable contribution to the community. The three presented methods to measure grain-size distributions in the field are commonly applied in other studies. The methods are well explained and carefully performed in the field. The paper is clearly written and I recommend to publish the manuscript in ESurf. However, I have some comments regarding the statistical analyses, the presentation of the data, the structure of the manuscript and the extend of the discussion.

The aim of the manuscript is to compare grain-size distributions. As such, statistical tests to investigate if distributions are different from each other or indistinguishable, are mandatory. One example is on page 5 line 28, where the authors state that the grain-size distribution in the surface layer is indistinguishable from the layers below. This statement needs to be supported with a statistical test. Another example is on page 7 line 1, where the authors report that above a threshold of 10000 kg the D50 and D90 are equivalent to the whole trench. I think that the identification of such a threshold should be based on statistical analyses. Calculating a moving mean and the according standard deviations and test when means become indistinguishable is one option. In addition, the measured grain-size distributions are only presented as cumulative density functions (CDF) in fig. 6. When plotted as CDF, differences in distributions are hard to detect by eye. For better comparison of the distributions, the probability density functions (PDF) and quantile or percentile plots should be added.

I think the structure of the manuscript is lacking a clear separation between the Methods, Results and Discussion sections. The Method section should be a clear description of the applied techniques, but should not contain any references to measured data. The Results section should be a neutral description of the data without any interpretation of it. I advise the authors to carefully check the manuscript and clearly separate method description, results description and interpretation. Below, I have listed a few points where the mixing was obvious to me:

- p. 3 lines 11-15: To me, these sentences belong to the Results, Discussion and Conclusion section.

- p. 4 lines 4 – 16: This paragraph mixes the methodological descriptions and results. The description and reference to Fig. 5 is part of the results section.

- p. 5 line 10: Same here, the reference to Fig. 5b belongs to the Results.

- p. 6 line 6-8: This is more than just the description of the result, and should be moved to the discussion.

- p. 6 lines 16-21: From my perspective, this entire paragraph belongs to the discussion section.

- p. 7 lines 2-3: The last sentence of this paragraph is discussion and not a description of the results.

- p. 7 lines: 8 – 11: These sentences belong to the discussion.

The authors clearly state in the Introduction that they test two hypotheses, namely the investigation of granulometric uniformity within the active layer and the application of surface point counts developed for horizontal layers on vertical layers. And both of these hypotheses are discussed later. However, the authors perform three different grain-size analyses. Currently, to verify or falsify their hypotheses, they only discuss two of them in detail, which are the volumetric analysis and the surface analysis on a vertical section in the trench. I think the paper would benefit from expanding the discussion about the reach-scale surface counts. As the authors state in their manuscript, vertical surface analyses are applied in paleo-studies. But these measurements are often compared to modern channel measurements, in which case a vertical surface count is compared to a horizontal surface count. In their study, the authors show that horizontal reach-scale surface count results in a coarser distribution than the vertical surface count from within the trench (Fig. 6d). Why is that? And what implication does this observation have for field studies that compare vertical with horizontal (or paleo and modern) grain-size distributions? I think it would be a missed opportunity to not extend the discussion (and maybe add a third hypothesis accordingly). However, if the authors decide to not include it, the third method (horizontal clast counts) can be removed from the paper.

An important point of the paper is that the investigated gravel-bed river has no armour layer and thus, any conclusion drawn from the findings are restricted to non-armoured channels. This restriction is mentioned in some parts of the paper, but not consistently throughout. From my point of view, this restriction needs to be mentioned in the abstract and potentially even in the title of the manuscript. Further clarifications of this restriction needs to be added to the sentence page 6 lines 9 – 11 and in the Conclusion (page 8 lines 27 - 31), which are currently phrased too generalized.

The abstract is currently fairly short. As an abstract serves as a stand-alone summary of a paper, the abstract could be extended by clearly listing the two hypotheses, the results and the according conclusions.

An important difference between this method-testing study and an applied study is that the analyses in this study are performed on a modern and active channel-bed. In paleo-studies, the vertical grain size measurements are applied to deposits that are thousands, sometimes millions of years old. I think it would be useful to mention within section 5.2 (page 8 lines 1 - 19), that the grain-size distributions in sedimentary deposits can also be altered after their deposition/ abandonment. Desert pavements, for example, can form in arid or semi-arid environments. Aeolian processes form a

coarse gravel layer of interlocked clasts at the surface, underlain by a layer of very fine material [e.g. McFadden et al., 1998]. Processes like this should be taken into account when applying the vertical sampling strategy to paleo-deposits. Other examples of post-depositional alterations include soil-production or bioturbation.

McFadden, L. D., E. V McDonald, S. G. Wells, K. Anderson, J. Quade, and S. L. Forman (1998), The vesicular layer and carbonate collars of desert soils and pavements: formation, age and relation to climate change, Geomorphology, 24, 101–145.

As this study compares different approaches and analyses, and aims to improve the reliability of characterizing grain-size distributions in the field, it would benefit from including the raw data of the field measurements as a supplementary file. That allows the re-analysis of the data for future studies.

The following points are minor comments only:

p. 2 line 8: "...at a reach scale..." I think this sentence needs some further explanation, maybe include rough dimensions or explain the term 'reach'.

p. 2 line 20: Please clarify in the second part of the sentence that the thickness of the active layer corresponds to the maximum elevation difference within a cross section and not in the downstream direction.

p. 3 line 6: D'Arcy et al. did not sample a vertical section, but the grain size distribution on the surface of an alluvial fan. Same accounts for p. 8 line 7.

p. 4 lines 26-27: List all sieve sizes used for the analysis, not only the minimum and maximum, since the size step can potentially affect the resolution of the datasets.

p. 5 line 27: Remove the extra comma and space after 0.2 m.

p. 6 line 9: Maybe clarify in the title that you compare the volumetric analysis to the horizontal surface counting and not the vertical surface counting.

p. 7 line 6: The vertical counts are not only shown in fig. 6d, but in all four graphs of

fig. 6.

p. 8 line 22: For clarification the authors could add 'horizontal and vertical' surface counts and volumetric samplings,. . .

p. 8 lines 20: As it is written now, the following paragraph is more a summary than a conclusion, so I suggest to adjust the title of this paragraph.

Fig. 4. It would help the reader to add length information to the pictures a, c and d.

Fig 5. Although stated in the figure caption, the figure does not really show uncertainties, but rather variability. How is the inherent variability defined? It would help to explain this at least in the figure caption. The combination of red and green colors (fig. 5b) is invisible for everybody suffering from red-green blindness.

Fig. 7 (caption). 'dashed' line instead of 'dotted'.

---

## Referee Comment (RC2) · Anonymous Referee #2 · 23 Jul 2018

This paper presents a field study in which the authors sample the surface and sub-surface sediment in an active braided gravel-bed river in a variety of ways, and compare the resulting grain size distributions. The overall intent of the work is to assess whether grain-size distributions collected from vertical exposures are representative of the overall grain-size distribution of the river bed, as this has important implications when interpreting data from outcrops in paleohydrology studies.

Overall this is a clear paper that presents useful data that should be of interest to the readers of Earth Surface Dynamics. There are, however, a few areas in which the

manuscript may be improved.

Much of the analysis relies on comparison of grain size distributions – either individual volumetric samples compared against each other, horizontally or vertically aggregated volumetric samples compared with each other, volumetric samples compared to Wolman-style point counts, surface transects compared to trench samples, and so on... – but the presentation limits the comparison to the D50 and D90, with some estimate of the uncertainty in each parameter, and visual comparison of cumulative grain-size distributions. Some sort of more rigorous statistical testing would greatly improve the main thrust of the paper. Some possible options could be the Mann-Whitney test to compare medians, or the Kolmogorov-Smirnov test to compare entire distributions.

Along these same lines, the grain size distributions are shown in Figure 6 with an arithmetic horizontal (grain size) axis. In some circumstances this may be okay, but in general with a wide range of grain sizes, as is the case here, it is preferable to use a logarithmic horizontal axis as it does not overly compress the finer range of grain sizes. Replotting the distributions with a logarithmic axis will also probably better represent how the D50 differs from one distribution to the next.

In addition to the D50 and D90, it would be instructive to see how the variability (perhaps quantified by the geometric standard deviation) of the grain size distributions varies as a function of the individual volumetric samples, and as samples are aggregated. I suspect the standard deviation of the individual samples is smaller than the aggregated samples, supporting the idea that individual morphologic features within the active layer are better sorted patches of sediment than the distribution of the active layer as a whole.

The results from the transects (the surface samples) are not really presented in the Results section of the paper. Currently they are mentioned only in passing in Section 4.3 and shown in Figure 6d. It would help to provide more information on these samples in the Results, and perhaps to add a table or amend a current table to include the

relevant grain size statistics from this dataset. Looking at Figures 5 and 6, it is not clear to me that the D50 of the surface transects and the D50 of the trench sediment are the same.

The Discussion section 5.2 on vertical sampling could be expanded to provide some more context to relate the present work to the stratigraphic record. An important outcome of the sampling strategy employed in the present study is that only the active layer (defined as ∼10*D90 thick) was sampled, and the authors conclude that if the sample size is large enough the grain size distribution does not vary in space throughout the active layer. In the rock record, deposits from different time periods are likely to have different active layer thicknesses, and these may be further changed after emplacement by erosion events, which may reduce the thickness of or even completely destroy an active layer. Some further discussion about how the findings in this paper may apply to paleo studies would be welcome.

Some other comments, by line number:

P. 4, line 27, and elsewhere: the word "weight" appears in several places in the manuscript, when it should be a different form of the word (i.e., here, it should be "We then weigh the grains. . .")

P. 7, line 3: "excesses" should be "exceeds". Also, what is the "typical size of the morpho-sedimentary elements of the bed"? Those data were not presented, and no mention of how to estimate them is given.

Figure 3: This figure could use a legend. And the vertical axis has no units? My interpretation of the plot is that the vertical axis is the deviation from the mean bed elevation at each cross section, which should still have units of (probably) meters.

Figure 5: How is "inherent variability" determined here? Fit by eye, or some statistical method?

Figure 8: Caption should say "Photographs", not "Photographies"

**ESurfD**

Interactive
comment

---

## Editor Comment (EC1) · R. Hodge (Editor) · 25 Jul 2018

We have now received two thorough reviews of your paper, which are both supportive of the paper. The reviewers agree that the question of whether grain size distributions (GSD) from vertical outcrops are equivalent to those measured from horizontal surface is a useful question to ask, and has implications for interpreting paleohydrology. The reviewers do, however, also provide you with many useful comments on how you can improve aspects of this paper.

[Figure]

One potential weakness that is identified by both reviewers is that you could use more statistical analysis to justify your interpretations. The two reviewers both give useful suggestions as to how this could be done, and I would encourage you to take these on board. Reviewer 2 also suggests that you could consider more the entire GSD, rather than just certain percentiles of it.

Given that a justification of this work is the application to sedimentary deposits, both reviewers observe that you could say more in the discussion about the processes that can affect the GSD of sedimentary deposits post-deposition, and more broadly consider the differences between this study of modern sediments and application to paleo deposits.

Reviewer 1 also suggests that the paper would benefit from some restructuring, and consideration of how the three different grain size analyses map onto the two identified hypotheses.

Best wishes, Rebecca Hodge

---

## Author Comment (AC1) · 31 Aug 2018

set

**L. Guerit et al.**

laure.guerit@get.omp.eu

**ASSOCIATED EDITOR R. Hodge (Editor) rebecca.hodge@durham.ac.uk**

**We have now received two thorough reviews of your paper, which are both supportive of the paper. The reviewers agree that the question of whether grain size distributions (GSD) from vertical outcrops are equivalent to those measured from horizontal surface is a useful question to ask, and has implications for interpret-**

[Figure]

ing paleohydrology. The reviewers do, however, also provide you with many useful comments on how you can improve aspects of this paper.

One potential weakness that is identified by both reviewers is that you could use more statistical analysis to justify your interpretations. The two reviewers both give useful suggestions as to how this could be done, and I would encourage you to take these on board. Reviewer 2 also suggests that you could consider more the entire GSD, rather than just certain percentiles of it.

Given that a justification of this work is the application to sedimentary deposits, both reviewers observe that you could say more in the discussion about the processes that can affect the GSD of sedimentary deposits post-deposition, and more broadly consider the differences between this study of modern sediments and application to paleo deposits.

Reviewer 1 also suggests that the paper would benefit from some restructuring, and consideration of how the three different grain size analyses map onto the two identified hypotheses.

Best wishes, Rebecca Hodge

First of all, we want to thank the reviewers and the associated editor for their careful work on our manuscript and their constructive comments. Following their suggestions, we have improved the manuscript and the main improvements can be summarized as follow:

- The paper is now based on samples issued from the trench only

- We add quantile-quantile plots to show that all the samples follow a lognormal distribution and we compare in more detail the grain-size distributions. In addition, we use ANOVA tests to support the similarity of the grain-size distributions of the five layers on one hand, and of the six columns on the other hand.

- The application of our study to ancient systems is discussed more thoroughly.

- We add three new figures, one table and the raw data as Supplementary Material

- The manuscript has been restructured to better separate the methodology, the results and the discussion.

We answer point-by-point to each comments of the reviewers.

**ANONYMOUS REFEREE #1**

**Guerit et al. present a field study from the Urumqi River, China in which they compare different methods of grain-size analysis including (i) horizontal surface counts over the whole river width, (ii) vertical surface counts on an outcropping trench wall and (iii) volumetric counts (sieving) of a 1 m deep trench excavated within the dry channel-bed. As they found no differences in sub-sample grain-size distributions in vertical nor horizontal direction within the trench, they propose that the grain-size distribution is uniform within the active layer, which might be a typical phenomenon for non-armoured gravel-bed, braided rivers. Second, they found no difference between the volumetric grain-size analysis and the vertical surface counts within the same trench. They conclude that the surface point count method, which was originally developed by Wolman for horizontal surface granulometry analyses in active rivers, can also be applied to vertical outcrops.**

**Temporal variations in grain-size distribution are used to reconstruct paleo environmental conditions including climate and tectonics. As such, it is important to investigate the differences between methods that are commonly applied to characterize grain-size distributions. As this study performs a very systematic**

**comparison of three of those methods in a natural, gravel-bed, braided river, and we generally lack those systematic method validations, the study is a valuable contribution to the community. The three presented methods to measure grain-size distributions in the field are commonly applied in other studies. The methods are well explained and carefully performed in the field. The paper is clearly written and I recommend to publish the manuscript in ESurf. However, I have some comments regarding the statistical analyses, the presentation of the data, the structure of the manuscript and the extend of the discussion.**

**The aim of the manuscript is to compare grain-size distributions. As such, statistical tests to investigate if distributions are different from each other or indistinguishable, are mandatory. One example is on page 5 line 28, where the authors state that the grain-size distribution in the surface layer is indistinguishable from the layers below. This statement needs to be supported with a statistical test.**

In the revised manuscript, we describe in more details the distributions to better support their similarities. In particular, we now present quantile-quantile plots for all the samples, showing that after being normalized by the $\phi$-scale (log2 based), they all follow a normal distribution (insets in Figure 5). The means and the standard deviations of the normalized samples are presented in Figure S1 ans Table S1. The individual volumetric samples are unfortunately badly designed for statistical analysis as they correspond to a weight for a given diameter, and not to a distribution of individual measurements. Accordingly, for these samples, we propose a visual analysis of the curves together with the comparison of the characteristic diameters (D50 and D90) to discuss the differences and similarities of these samples. However, as our normalized samples follow a normal distribution, ANOVA tests are well designed to determine of the median diameters of the grain-size distributions of the five layers, or of the six columns, are similar or not. The two ANOVA tests added the revised version confirm the uniformity in grain-size at the scale of the active layer (Table 3). This approach is fully described in the Methodology section (p. 5, l. 8-22).

**Another example is on page 7 line 1, where the authors report that above a threshold of 10000 kg the D50 and D90 are equivalent to the whole trench. I think that the identification of such a threshold should be based on statistical analyses. Calculating a moving mean and the according standard deviations and test when means become indistinguishable is one option.**

As discussed above, the individual volumetric samples are not well-designed for statistical analysis and we choose to adapt the bootstrap method to evaluate the variation of the characteristic diameters (the D50 and the D90) with the sample weight (Figure 7). To built this figure, we randomly merge without replacement 1 to 30 of the volumetric samples and we determine the D50 and the D90 of 600 composite distributions. Then, we visually determine the weight of the distributions showing a D50 and a D90 similar to the bulk distribution issued from the trench, within the same confidence interval (i.e. +/- 5%). A statistical analysis is not required for such analysis. Accordingly, we only slightly modify the method and the figure, and explain it in more details in the revised manuscript (p. 7, l. 21-32).

**In addition, the measured grain-size distributions are only presented as cumulative density functions (CDF) in fig. 6. When plotted as CDF, differences in distributions are hard to detect by eye. For better comparison of the distributions, the probability density functions (PDF) and quantile or percentile plots should be added.**

Grain-size studies are generally based on some characteristic diameters that correspond to a given quantile of the grain-size distribution (i.e. the D50 is the 50th quantile of the distribution). CDF plots allow a direct read of the diameter associated to any quantile of the distribution and we therefore favor these plots instead of PDFs. Nevertheless, following the suggestion of Reviewer 2, we now present Figure 5 in logarithmic scale for the diameters, and the differences between the curves are now easier to read. In addition, we include quantile-quantile plots for the individual and bulk volumetric samples and for the vertical surface sample to illustrate that all the $\phi$-normalized

samples follow a normal distribution (insets Figure 5), and the mean and the standard deviation of the normalized distributions are presented on Figure S1 and Table S1. We also describe in a more systematic manner the differences and the similarities between the curves in the Results section.

**I think the structure of the manuscript is lacking a clear separation between the Methods, Results and Discussion sections. The Method section should be a clear description of the applied techniques, but should not contain any references to measured data. The Results section should be a neutral description of the data without any interpretation of it. I advise the authors to carefully check the manuscript and clearly separate method description, results description and interpretation. Below, I have listed a few points where the mixing was obvious to me: p. 3 lines 11-15: To me, these sentences belong to the Results, Discussion and Conclusion section. p. 4 lines 4 - 16: This paragraph mixes the methodological descriptions and results. The description and reference to Fig. 5 is part of the results section. p. 5 line 10: Same here, the reference to Fig. 5b belongs to the Results. p. 6 line 6-8: This is more than just the description of the result, and should be moved to the discussion. p. 6 lines 16-21: From my perspective, this entire paragraph belongs to the discussion section. p. 7 lines 2-3: The last sentence of this paragraph is discussion and not a description of the results. p. 7 lines: 8 -11: These sentences belong to the discussion.**

We reconsider the global organization of the manuscript to better separate the method from the results, and the results from the discussion. In particular, sentences related to Figure 5 have been moved to the Results section and the paragraph about armouring to the Discussion (p. 8, l. 1-7). However, we believe that the results of our analysis should be written explicitly. Therefore, we did not remove the last sentences of the Results subsections.

**The authors clearly state in the Introduction that they test two hypotheses, namely the investigation of granulometric uniformity within the active layer and**

**ESurfD**
**the application of surface point counts developed for horizontal layers on vertical layers. And both of these hypotheses are discussed later. However, the authors perform three different grain-size analyses. Currently, to verify or falsify their hypotheses, they only discuss two of them in detail, which are the volumetric analysis and the surface analysis on a vertical section in the trench. I think the paper would benefit from expanding the discussion about the reach-scale surface counts. As the authors state in their manuscript, vertical surface analyses are applied in paleo-studies. But these measurements are often compared to modern channel measurements, in which case a vertical surface count is compared to a horizontal surface count. In their study, the authors show that horizontal reach-scale surface count results in a coarser distribution than the vertical surface count from within the trench (Fig. 6d). Why is that? And what implication does this observation have for field studies that compare vertical with horizontal (or paleo and modern) grain-size distributions? I think it would be a missed opportunity to not extend the discussion (and maybe add a third hypothesis accordingly). However, if the authors decide to not include it, the third method (horizontal clast counts) can be removed from the paper.**

The equivalence between the horizontal surface count and the volumetric methods has been studied by previous workers and the two approaches lead to similar and directly comparable grain-size distributions (as mentioned in the Introduction (p. 3, l. 3-4), Church et al, 1987; Bunte and Abt, 2001). Therefore, in the revised version, we focus on the equivalency between the vertical count and the volumetric method only. This is also motivated by an issue with the data acquisition: the granulometric study of the trench was performed in 2008 whereas the surface sample was acquired during another field campaign, in 2010. Unfortunately, these two years appear to be the driest (2008) and the wettest (2010) years of the decade (see attached figure). We therefore suspect that the difference between the distributions arises from this change in water flux. However, we don't have data to support (or to reject) this idea and thus, following the suggestion of the reviewer, we decide to remove the horizontal surface count from

the revised version as it is not required for the main purpose of our work.

**An important point of the paper is that the investigated gravel-bed river has no armour layer and thus, any conclusion drawn from the findings are restricted to non-armoured channels. This restriction is mentioned in some parts of the paper, but not consistently throughout. From my point of view, this restriction needs to be mentioned in the abstract and potentially even in the title of the manuscript. Further clarifications of this restriction needs to be added to the sentence page 6 lines 9 - 11 and in the Conclusion (page 8 lines 27 - 31), which are currently phrased too generalized.**

We now explicitly write that the methods are equivalent on the Urumqi River bed (p. 7, l. 6) and mention in the discussion section that it might be the case for any non-armoured rivers(p. 8, l. 13) . We also add "in non-armoured, gravel-bed rivers" in the Conclusions (p. 9, l. 11).

**The abstract is currently fairly short. As an abstract serves as a stand-alone summary of a paper, the abstract could be extended by clearly listing the two hypotheses, the results and the according conclusions.**

In the revised abstract, we now present the two ideas tested in this work and write explicitly that vertical counts can be used to accurately sample grain-size distributions of paleo-braided rivers (p. 1, l.3-4 and 10-13).

**An important difference between this method-testing study and an applied study is that the analyses in this study are performed on a modern and active channel-bed. In paleo-studies, the vertical grain size measurements are applied to deposits that are thousands, sometimes millions of years old. I think it would be useful to mention within section 5.2 (page 8 lines 1 - 19), that the grain-size distributions in sedimentary deposits can also be altered after their deposition/ abandonment. Desert pavements, for example, can form in arid or semi-arid environments. Aeolian processes form a coarse gravel layer of interlocked clasts at**

**the surface, underlain by a layer of very fine material [e.g. McFadden et al., 1998]. Processes like this should be taken into account when applying the vertical sampling strategy to paleo-deposits. Other examples of post-depositional alterations include soil-production or bioturbation. McFadden, L. D., E. V McDonald, S. G. Wells, K. Anderson, J. Quade, and S. L. For- man (1998), The vesicular layer and carbonate collars of desert soils and pavements: formation, age and relation to climate change, Geomorphology, 24, 101-145.**

We ass a paragraph dedicated to the evolution of gravel sediments after deposition in the revised version. We now discuss that deposits can be affected by several processes such as wind deposits, soil development, or chemical alteration and we propose some methodological considerations to face these secondary processes (p. 8, l. 21-30).

**As this study compares different approaches and analyses, and aims to improve the reliability of characterizing grain-size distributions in the field, it would benefit from including the raw data of the field measurements as a supplementary file. That allows the re-analysis of the data for future studies.**

The dataset is now available as a Supplement.

**The following points are minor comments only:**

**p. 2 line 8: ". . .at a reach scale. . ." I think this sentence needs some further explanation, maybe include rough dimensions or explain the term 'reach'.**

We clarify the term reach by the addition of "(i.e. at the scale of the whole river bed, from several dozens to several hundreds of meters)" (p. 2, l. 12-13).

**p. 2 line 20: Please clarify in the second part of the sentence that the thickness of the active layer corresponds to the maximum elevation difference within a cross section and not in the downstream direction.**

corrected (p. 2, l. 24-25).

**p. 3 line 6: D'Arcy et al. did not sample a vertical section, but the grain size distribution on the surface of an alluvial fan. Same accounts for p. 8 line 7.**

This reference has been removed.

**p. 4 lines 26-27: List all sieve sizes used for the analysis, not only the minimum and maximum, since the size step can potentially affect the resolution of the datasets.**

We now write "We sieve the sediments using mesh sizes ranging from 63 $\mu$m up to 25.6 cm. Each mesh size is twice the previous one and we add three sieves (24, 48 and 96 mm) to obtain a more detail description in the gravel range." (p. 4, l. 16-18).

**p. 5 line 27: Remove the extra comma and space after 0.2 m.**

corrected (p. 6, l. 10).

**p. 6 line 9: Maybe clarify in the title that you compare the volumetric analysis to the horizontal surface counting and not the vertical surface counting.**

This subsection has been removed from the revised version.

**p. 7 line 6: The vertical counts are not only shown in fig. 6d, but in all four graphs of fig. 6.**

corrected (p. 7, l. 3-4).

**p. 8 line 22: For clarification the authors could add 'horizontal and vertical' surface counts and volumetric samplings.**

The horizontal surface sample has been removed from the revised version.

**p. 8 lines 20: As it is written now, the following paragraph is more a summary than a conclusion, so I suggest to adjust the title of this paragraph.**

This final paragraph proposes a summary of our work together with the conclusions drawn from our results. We therefore believe that the title (which corresponds to the

journal's format) is appropriate.

**Fig. 4. It would help the reader to add length information to the pictures a, c and d.**

This figure has been modified and picture a has been removed. For pictures c and d (b and c in the revised version), the perspective of the pictures prevents to propose a relevant scale so we now indicate the dimensions of the trench in the caption.

**Fig 5. Although stated in the figure caption, the figure does not really show uncertainties, but rather variability. How is the inherent variability defined? It would help to explain this at least in the figure caption. The combination of red and green colors (fig. 5b) is invisible for everybody suffering from red-green blindness.**

Indeed, this figure did not show uncertainties, but a range of variability around a mean value derived from a bootstrap analysis. This variability corresponds to the confidence interval (previously named the inherent variability) of the analyzed parameters (namely, the D50 and D90) for a given sample size. We modify the figure, the caption and the related text to explicite this approach (p. 5, l. 6-7 and p. 7, l. 21-25). The color issue was corrected.

**Fig. 7 (caption). 'dashed' line instead of 'dotted'. .**

corrected.

**ANONYMOUS REFEREE 2**

**This paper presents a field study in which the authors sample the surface and sub- surface sediment in an active braided gravel-bed river in a variety of ways, and com- pare the resulting grain size distributions. The overall intent of the work is to assess whether grain-size distributions collected from vertical expo-**

**sures are representative of the overall grain-size distribution of the river bed, as this has important implications when interpreting data from outcrops in paleo-hydrology studies.**

**Overall this is a clear paper that presents useful data that should be of interest to the readers of Earth Surface Dynamics. There are, however, a few areas in which the manuscript may be improved.**

**Much of the analysis relies on comparison of grain size distributions - either individual volumetric samples compared against each other, horizontally or vertically aggregated volumetric samples compared with each other, volumetric samples compared to Wolman-style point counts, surface transects compared to trench samples, and so on. . . - but the presentation limits the comparison to the D50 and D90, with some estimate of the uncertainty in each parameter, and visual comparison of cumulative grain- size distributions. Some sort of more rigorous statistical testing would greatly improve the main thrust of the paper. Some possible options could be the Mann-Whitney test to compare medians, or the Kolmogorov-Smirnov test to compare entire distributions.**

The volumetric samples are unfortunately badly designed for statistical analysis as they correspond to a weight for a given diameter, and not to a distribution of individual measurements. Accordingly, for these samples, a visual analysis of the curves together with the comparison of the characteristic diameters (such as the D50 and D90) appear as the best approach to discuss the differences and similarities of these samples. To support the visual comparison of the curves, we add QQ plots to the revised version showing that after being normalized by the $\phi$-scale (log2 based), all the samples follow a lognormal distribution (insets Figure 5). However, it is possible to use a statistical approach to compare the grain-size distributions of the five layers, or of the six columns, as the layers or columns form different groups described by several samples. As our normalized samples follow a normal distribution, we can use parametric tests and ANOVA tests appear as the best approach to determine whether the median diameters of the grain-size distributions of the five layers, or of the six columns, are similar or not. The two ANOVA tests added the revised version confirm the uniformity in grain-size at the scale of the active layer (Table 3). This approach is fully described in the Methodology section (p. 5, l. 8-22).

**Along these same lines, the grain size distributions are shown in Figure 6 with an arithmetic horizontal (grain size) axis. In some circumstances this may be okay, but in general with a wide range of grain sizes, as is the case here, it is preferable to use a logarithmic horizontal axis as it does not overly compress the finer range of grain sizes. Replotting the distributions with a logarithmic axis will also probably better represent how the D50 differs from one distribution to the next.**

We agree that differences between distributions are easier to read from logarithmic plots and we modified Figure 5 accordingly.

**In addition to the D50 and D90, it would be instructive to see how the variability (perhaps quantified by the geometric standard deviation) of the grain size distributions varies as a function of the individual volumetric samples, and as samples are aggregated. I suspect the standard deviation of the individual samples is smaller than the aggregated samples, supporting the idea that individual morphologic features within the active layer are better sorted patches of sediment than the distribution of the active layer as a whole.**

As discussed above, the volumetric samples are not designed for statistical analysis and accordingly, on Figure 7, we propose a visual estimation of the variation of two characteristic diameters (the D50 and the D90) with the sample weight. This figure is built by artificially and randomly merging without replacement the 30 volumetric samples issued from the trench, and we observe that light samples show a greater variability around the mean values than larger samples. This illustrates that the variability in grain-size distributions observed at small scale (i.e. at the size of our volumetric

samples) is a local feature that vanishes at the scale of the whole river bed. This point is discussed more carefully in the revised manuscript (p. 7, l. 21-32).

**The results from the transects (the surface samples) are not really presented in the Results section of the paper. Currently they are mentioned only in passing in Section 4.3 and shown in Figure 6d. It would help to provide more information on these samples in the Results, and perhaps to add a table or amend a current table to include the relevant grain size statistics from this dataset. Looking at Figures 5 and 6, it is not clear to me that the D50 of the surface transects and the D50 of the trench sediment are the same.**

As discussed in the answer to the first reviewer, the horizontal surface sample has been removed from the revised version and we now focus on the similarity between the samples issued from the trench only, i.e. the volumetric samples and the vertical surface count.

**The Discussion section 5.2 on vertical sampling could be expanded to provide some more context to relate the present work to the stratigraphic record. An important out- come of the sampling strategy employed in the present study is that only the active layer (defined as âĹij10\*D90 thick) was sampled, and the authors conclude that if the sample size is large enough the grain size distribution does not vary in space throughout the active layer. In the rock record, deposits from different time periods are likely to have different active layer thicknesses, and these may be further changed after emplacement by erosion events, which may reduce the thickness of or even completely destroy an active layer. Some further discussion about how the findings in this paper may apply to paleo studies would be welcome.**

This section has been extended in the revised version. In particular, we now explicitly write that the absolute thickness of the active layer may vary in time and that deposits can be eroded (p. 8, l. 32-35).

**Some other comments, by line number:**

**P. 4, line 27, and elsewhere: the word "weight" appears in several places in the manuscript, when it should be a different form of the word (i.e., here, it should be "We then weigh the grains. . ." .**

This has been corrected throughout the manuscript.

**P. 7, line 3: "excesses" should be "exceeds". Also, what is the "typical size of the morpho-sedimentary elements of the bed"? Those data were not presented, and no mention of how to estimate them is given.**

These sentences have been changed in the revised version but the spatial scale of the morpho-sedimentary elements is now explicitly mentioned (p. 7, l. 19)

**Figure 3: This figure could use a legend. And the vertical axis has no units? My interpretation of the plot is that the vertical axis is the deviation from the mean bed elevation at each cross section, which should still have units of (probably) meters.**

We add units to the vertical axis (indeed, it is in meter) and add to the caption "Elevation is given as the deviation from the mean bed elevation at each cross section."

**Figure 5: How is "inherent variability" determined here? Fit by eye, or some statistical method?**

Following this comment and the one from Reviewer 1, we modify the figure, the caption and the text to better explain the boostrap approach used here. The confidence interval (previously named inherent variability) is defined as the variability around the mean values, for a given sample size (p. 5, l. 6-7 and p. 7, l. 21-25).

**Figure 8: Caption should say "Photographs", not "Photographies" .**

corrected (this figure has been moved to Supplementary material)

**ESurfD**

Interactive
comment

**REFERENCES**

Yao, J., Zhao, Y., and Yu, X. (2018). Spatial-temporal variation and impacts of drought in Xinjiang (Northwest China) during 1961-2015. PeerJ, 6, e4926.

**ESurfD**
[Figure]

[Figure]

**Fig. 1.** Average precipitation in Xinjiang between 2005 and 2015 (data from Yao et al, 2018)

---

## Author Response (AR2)

**Comments by Rebecca Hodges**

**Thanks for addressing the comments made by the two reviewers. The changes that you have made have improved the manuscript and I think that it is almost ready for publication. In the attached file I have identified a small number of places where the manuscript would benefit from further clarification, and ask you to address these changes.**

The manuscript has been improved and clarified according to your comments and suggestions. Minor points have been corrected and you'll find below our point-by-point answer to the main comments, together with the document highlighting the differences between this version and the previous one.

**Page 2: Specify what you mean by 'this'**
We now write "For such reconstructions,"

**Page 3: Lines 4 to 12 need some clarification. You talk about two methods (photo sieving and the surface count) and two sediments (cemented and uncemented), and it's not clear to me which application you need to test - is it the application of surface counts to cemented grains, or surface counts to vertical surfaces of uncemented grains? It seems to me that if you have a cemented outcrop then there are issues relating to grain exposure, regardless of whether you are photoseiving or using a surface count method.**
**As an aside, when using photosieving of surface samples, I would tend to assume that the two axes measured in the photo are the a and b axis. If looking at a vertical section, do you have to assume that they are the a or b, and c, axes instead?**
Indeed, issues related to cemented sediments are not related to the method. We rephrase this sentence to clarify why we suggest that photographic methods are not ideal and we now write "but grain identification is strongly affected by the shadows and lights on the picture or by the resolution of the camera, and these methods often require a non-negligible amount of operator corrections to accurately measure the diameters".

In fact, a- and b-axis should be visible at the surface of a river bed, whereas in depth, the visible diameters depend upon the orientation of the cross-section with respect to the (paleo-)flow. Our analysis is based on uncemented sediments so that we were able to remove grains and to identify the three axis. The identification of the b-axis was thus very easy, but it can be an issue for cemented outcrops. This question is addressed in the Discussion.

**Page 4: Methods should be in the past tense, as it is what you have already done.**
corrected

**Page 5: It's confusing to me that you say that the volumetric samples are not for statistical analysis, and then go on to say how you conducted statistical analysis of the data! Suggest rephrasing the first sentence (line 30).**
Statistical tests can be used to compare two distributions to determine whether the difference in means and standard deviation are significant or not. However, such tests (like the ones suggested by the reviewers) are based on the number of measurements. Here, we don't have access to the number of grains of a given diameter, but only to the proportion of grain of a given diameter in the

total distribution. Accordingly, we can not compare statistically the distributions of the individual samples. Then, when the samples are merged with respect to their position in the trench, they can be used as several samples from the same group. For example, the layer 1 is characterized by five samples. We thus have several groups characterized by several samples. In such configuration, we can use ANOVA tests to compare the variability within the groups to the variability between the groups. We rephrase the paragraph to clarify our approach.

**Page 6: Suggest rephrasing - you first say that they are very similar, but then say that there is some scatter.**

We now write " In addition, the means and the standard deviations of the fitted curves are in the same range of values (Fig. S1, Table S1), and in consequence, the distributions show a similar shape and plot close to each other "

**Page 6: Isn't this paragraph results?**

Our main result is that the grain-size distribution is homogeneous at the scale of the active layer. Here, we rather discuss the fact there is some variability at a scale smaller than the active layer, and we visually determine a minimum sample weight to get over this variability. We thus believe that this paragraph belongs to the discussion as it is an extra analysis performed to discuss the results. In consequence, we did not move this section.

**Figure 6: The D50 dots with white edges are hard to see - can you use another colour?**

We agree that this version was not very satisfying and we now propose to show the D50 in green and the D90 in gray.

**Tables 1, 2, 4 and 5: Do you mention this in the main text anywhere? If not, it might be useful to add it.**

In the Methodology section, we write that the uncertainties associated with our grain-size distributions are related to the weight of the largest stones with respect to the weight of the samples, after the criteria defined by Church et al (1987). To make the tables more clear, we add "(see Methodology for details)" in the captions.

[revised manuscript text omitted]